# A Miniaturized Loaded Open-Boundary Quad-Ridge Horn with a Stable Phase Center for Interferometric Direction-Finding Systems

**DOI:** 10.3390/mi16010044

**Published:** 2024-12-30

**Authors:** Zibin Weng, Chen Liang, Kaibin Xue, Ziming Lv, Xing Zhang

**Affiliations:** The National Key Laboratory of Radar Detection and Sensing, Xidian University, Xi’an 710071, China; liangchen@stu.xidian.edu.cn (C.L.); kbxue@stu.xidian.edu.cn (K.X.); 23021211267@stu.xidian.edu.cn (Z.L.); monicastarstar@163.com (X.Z.)

**Keywords:** common-mode current (CMC), stable phase center, radio interferometry, ultrawideband antennas

## Abstract

In order to achieve high accuracy in interferometric direction-finding systems, antennas with a stable phase center in the working bandwidth are required. This article proposes a miniaturized loaded open-boundary quad-ridge horn (LOQRH) antenna with dimensions of 40 mm × 40 mm × 49 mm. First, to stabilize the phase center of the antenna, the design builds on the foundation of a quad-ridge horn antenna, where measures such as optimizing the ridge structure and introducing resistive loading were implemented to achieve size reduction. Second, electrically small-sized antennas are more susceptible to the effects of common-mode currents (CMCs), which can reduce the symmetry of the radiation pattern and the stability of the phase center. To avoid the generation of common-mode currents during operation, a self-balanced feed structure was introduced into the proposed antenna design. This structure establishes a balanced circuit and routes the feedline at the voltage null point, effectively suppressing the common-mode current. As a result, the miniaturization of the LOQRH antenna was achieved while ensuring the suppression of the common-mode current, thereby maintaining the stability of the antenna’s electromagnetic performance. The measured results show that the miniaturized antenna has a small phase center change of less than 20.3 mm within 2–18 GHz, while the simulated phase center fluctuation is only 14.6 mm. In addition, when taking 18.5 mm in front of the antenna’s feed point as the phase center, the phase fluctuation is less than 22.5° within the required beam width. Along with the desired stable phase center, the miniaturized design makes the proposed antenna suitable for interferometric direction-finding systems.

## 1. Introduction

Accurate positioning and localization are essential in modern technologies such as radar detection, communication systems [1,2], and electronic warfare. Traditional methods like time of arrival (TOA) [3,4,5], frequency-based estimation, and velocity-based techniques have been widely studied for localization. While these methods have their merits, they often face challenges in complex environments, such as susceptibility to errors from signal timing, mobility, or the need for extensive hardware and signal processing.

In contrast, Direction of Arrival (DOA) estimation [6,7,8] has gained significant attention due to its ability to provide precise localization through the measurement of signal angles. Using multiple antennas, DOA estimation is highly effective for applications like satellite navigation, automotive systems, and communication networks.

Figure 1 shows the simplified interferometric model for DOA estimation, which ignores factors such as differences between receiving channels and the coupling between antenna elements.

The calculation process of the simplified interferometric model is shown in (1)–(3). Table 1 shows the detailed explanation of each parameter in Equations (1)–(3).
(1)$$\theta_{DOA}=\arcsin(\frac{\lambda }{2\pi }\sum_{k=1}^{N-1}d_{k}\phi_{W,k}/\sum_{k=1}^{N-1}d_{k}^{2})$$


(2)
$$\left\{\begin{array}{c} \phi_{W,k+1}=\phi_{Mea,k+1}+\gamma_{Mea,k+1}+\sigma \\ \gamma_{Mea,k}=2\pi \cdot floor(\frac{\alpha_{k}\phi_{W,k}}{2\pi }),\alpha_{k}=\frac{d_{k+1}}{d_{k}},\phi_{W,1}=\phi_{Mea,1} \end{array}\right.$$



(3)
$$\sigma =\left\{\begin{array}{cc} 0 & \phi_{Mea,k+1}+\gamma_{Mea,k+1}-\alpha_{k}\phi_{W,k}\in [-\pi ,\pi )\\ 2\pi & \phi_{Mea,k+1}+\gamma_{Mea,k+1}-\alpha_{k}\phi_{W,k}<-\pi \\ -2\pi & \phi_{Mea,k+1}+\gamma_{Mea,k+1}-\alpha_{k}\phi_{W,k}\geq \pi \end{array}\right.$$


As electromagnetic field detectors in the interferometer direction-finding system, the performance of antennas is crucial for receiving signals, particularly phase center stability. Since the DOA is calculated based on the phase difference between different channels and the baseline length (distance between unit antennas), any phase measurement error can significantly impact detection accuracy.

According to the reciprocity characteristic of the antenna, the phase center is also the location of the antenna reference point (ARP), where the antenna receives signals [9,10]. There is a unique phase center for an ideal antenna, but this is impossible in practice: different signal frequencies will cause phase center offset (PCO), and different signal incident angles also cause phase center variations (PCVs) [11,12]. Fluctuating phase centers will cause changes in the baseline of the interferometer, which will inevitably introduce deviations in the test results.

In the case of only considering the phase detection error Δϕ, which is caused by the fluctuation of the phase center, the baseline interferometry error $\Delta \theta_{i}$ can be given by Equation (4):(4)$$\Delta \theta_{i}=\frac{\lambda_{\min}\cdot \Delta \phi }{2\pi d_{i}\cos\theta_{DOA}}$$

Since the phase center of most antennas varies with azimuth and is challenging to solve analytically, there are two main methods to correct the errors caused by phase center fluctuations. The first method is to ensure consistency across the antenna units. When all units operating in the same frequency band are consistent, PCO and PCV will only cause a small shift in the baseline, which will not significantly affect the test results. The second method involves making sufficient corrections. If PCO and PCV fluctuations lead to large changes in the baseline length, repeated measurements are needed to identify the pattern, allowing for corrections to eliminate this error [13,14,15,16,17,18,19,20].

However, when antenna units are without choking [21,22] structures, a part of the feed current will flow on the outer surface of the coaxial cable, generating the common-mode current (CMC). For electrically large-sized antennas, the influence of the CMC is often ignored. However, the influence must be addressed for many electrically small-sized antennas. Impacted by the CMC, the feeding structures easily become a part of the radiation structure, largely affecting the radiation pattern symmetry and phase center stability. The CMC will distort the radiation characteristics of the antenna units, and its influence on the phase center is complex to evaluate [23,24]. In addition, there will inevitably be mutual coupling between the units, which will also make the PCO and PCV change irregularly and affect the measurement accuracy.

To address these challenges, this article proposes a miniaturized loaded open-boundary quad-ridge horn (LOQRH) antenna, designed to maintain stable phase center characteristics while minimizing the impact of CMC. The proposed antenna incorporates several key design techniques, including optimizing the ridge curve and incorporating resistive loading, which reduces surface current flow and achieves compact dimensions of 0.27$\lambda_{L}$ × 0.27$\lambda_{L}$ × 0.33$\lambda_{L}$ (where $\lambda_{L}$ is the free-space wavelength at the lowest operating frequency). Furthermore, by conducting an odd–even mode analysis [25,26], this article gives point N as the lead-out position for the coaxial feedline. When the feedline is led from point N, there is no energy flowing on the outer surface of the feedline, which also means no CMC. These key design techniques enhance phase center stability and improve radiation pattern symmetry, making the LOQRH antenna ideal for DOA estimation systems.

## 2. Antenna Geometry and Design

### 2.1. Structure of the Proposed Antenna

Figure 2a shows the whole 3D structure of the antenna unit, which includes three main components: the ridged horn, the loaded resistor, and the self-balance feeding structure.

The ridge is critical in the impedance transformation from the feed point to the aperture. The proposed antenna combines an additional Bessel curve with the exponential curve to optimize the ridge more flexibly.

L indicates the length of the exponential curve in the z direction, and $h_{r}$ indicates the length of the Bessel curve in the y direction. Y’(L), *g*, and *d* are the slope at the end of Y(z), the diameter of the aperture, and the ridge spacing, respectively. What is more, the smallest ridge spacing can lower the ridge waveguide’s cut-off frequency and make the equivalent impedance closer to 50 Ω.

As shown in Figure 2b, we perform 45° chamfering treatments on ridges, which prevents the radiation performance from being affected by too large ridge spacing and avoids the collision of ridges caused by too small ridge spacing.

To improve the low-frequency characteristics of the proposed LOQRH, a resistor is loaded between each ridge and the fixed flange to reduce the ridge length and absorb the energy not radiated. The VSWR performance of the antenna loaded with resistors of different resistance values and an unloaded resistor is given in Figure 3. It can be seen that the VSWR of the antenna is significantly decreased at 2–5 GHz after loading the resistor compared to when no resistor is loaded, with little change at high frequencies. Integrating the proposed antenna’s S_11_ and gain performance, the resistors are chosen as 510 Ω.

As shown in Figure 2c, two 047 50 Ω semi-rigid coaxial probes are used for feeding to facilitate processing and miniaturizing. The two probes are placed orthogonally for dual polarization and better isolation between the ports. For the VSWR of the ports to be as similar as possible, it is necessary to make the two probes as close as possible without contact. Finally, the distance between them is 0.5 mm.

There is also a stepped cavity with a shorting plane at the bottom of the ridge waveguide, as shown in Figure 2c. To make the shorting plane realize high impedance characteristics in the entire frequency band, the height $L_{2}$ and length W of the back cavity steps can be adjusted, thereby inhibiting the backward transmission of electromagnetic waves.

In addition, the coaxial feedline will be distributed along the claw structure and led out at the bottom center of the claw structure, as shown in Figure 2c. The detailed mechanism will be discussed later.

In summary, the proposed miniaturization design combines an exponential curve and a Bessel curve for the ridge profile, ensuring a smooth transition of the antenna’s characteristic impedance from the feed point to the bell mouth surface, with chamfered ridge edges for enhanced performance. Additionally, resistors are loaded at the ends of each ridge to absorb the energy that is not radiated due to the reduced vertical size of the ridge at low frequencies, effectively improving low-frequency performance. Compared to the traditional four-ridge horn antenna, the antenna proposed in this paper achieves a reduction in both the vertical dimension ***L*** and aperture size *g*. To attenuate higher-order modes in the horn, the opening section length is typically greater than 0.5λ (where λ corresponds to the lowest frequency). For 2 GHz, this value is 75 mm, whereas the proposed antenna achieves a length of 30 mm. Furthermore, as the vertical size *L* of the antenna decreases, the aperture size *g* also reduces. Therefore, the proposed four-ridge horn antenna successfully reduces both the vertical dimension *L* and aperture size *g*, leading to a compact design.

### 2.2. Common-Mode Current Suppression

The fluctuation of the antenna’s phase center is often related to the antenna’s dimensions and also weakens as the antenna’s dimensions decrease. Therefore, interferometric direction-finding antennas can stabilize their phase center once their dimensions are small enough. Also, benefitting from the limited dimensions, interferometric direction-finding antennas can reduce coupling between themselves. In this way, each unit can be a “similar point source”, which is essential for ensuring the antennas’ performance and improving interferometric direction-finding systems’ accuracy.

Due to the potential difference between the inner and outer surfaces of the outer conductor of the coaxial line, a part of the current on the inner surface flows back to the source along the outer surface. This part of the current is the CMC. The miniaturized antenna is more susceptible to the influence of the CMC. The CMC affects the feed balance, causing the pattern to deviate, intensifying phase center fluctuations, and even introducing unnecessary cross-polarization [27,28,29].

For the feeding structure, the inner conductor of the coaxial line is connected to a ridge, and the outer conductor is connected to the opposite ridge. To suppress the CMC, the proposed antenna added a claw structure after the shorting plane, as shown in Figure 2c. The coaxial feedline will be distributed along the claw structure and led out at the bottom center of the claw structure.

The self-balanced feeding structure’s equivalent model of the odd and even modes is shown in Figure 4. $A_{1}$ and $A_{2}\;$ are the input electric field at points $M_{1}$ and $M_{2}$, respectively. The electrical length from point $M_{1}$ to point N is $\theta_{1}$, and the electrical length from point $M_{1}$ to point $M_{2}$ is $\theta_{T}$. Then, the odd-mode input $A_{o}=(A_{1}-A_{2})/2$, and even-mode input $A_{e}=(A_{1}+{\;A}_{2})/2$. The function of electric field distribution and electric length can be expressed by $e^{-\alpha \theta }e^{j\beta \theta }$, where α represents the attenuation factor of the electric field as a function of electrical length, and β is the propagation factor. At point *N*, where the coaxial feedline is led out, the electric field B can be expressed by Equation (9):(5)$$B=(A_{e}+A_{o})\cdot e^{-\alpha \theta_{1}}\cdot e^{j\beta \theta_{1}}+(A_{e}-A_{o})\cdot e^{-\alpha (\theta_{T}-\theta_{1})}\cdot e^{j\beta (\theta_{T}-\theta_{1})}$$

Let *B* = 0, then
(6)$$\theta_{1}=\frac{1}{2}\theta_{T}+\frac{1}{2(-\alpha +j\beta )}\ln(-\frac{A_{1}}{A_{2}})$$

Near the midpoint, there must be a point where the electric field intensity is 0. When leading from this point, no energy flows on the outer layer of the coaxial line, thereby suppressing the occurrence of the CMC from its source. This point is taken as the lead-out point, point *N*.

Figure 5 shows the surface current distribution and the normalized radiation patterns at some frequency points. When feeding without CMC suppression, there will be a CMC on the coaxial line, and the pattern of the antenna will have obvious distortion accordingly; when feeding with CMC suppression, there will be no CMC on the coaxial line, and the pattern of the antenna also maintains good symmetry.

The specific normalized radiation pattern distortion is shown in the gray ovals in Figure 5, while the specific details of common-mode current suppression are shown in the red ovals in Figure 5.

At the same time, since the common-mode current suppression effect is frequency related, we show the effect of the CMC suppression structure at high frequencies in Figure 6. Combining Figure 5 and Figure 6, we can clearly see that for antennas with electrically small-sized antennas, the common-mode current has a large effect. Due to the effect of common-mode current, the radiation pattern is severely distorted. Thus, for our proposed LOQRH antenna, the effect of the common-mode current is much larger in the low-frequency case than in the high-frequency case.

Therefore, the CMC suppression structure can effectively suppress the CMC on the coaxial line and reduce the antenna pattern distortion caused by the CMC.

### 2.3. Phase Center Stability

This article solves the phase centers of the *XOZ* and *YOZ* planes, respectively, and then performs an arithmetic average on them to obtain the equivalent phase center in the entire space. As to the proposed antenna, it can be assumed that ($x_{0}$, $y_{0}$, $z_{0}$) is its phase center in a particular plane. We have simulated in CST to verify that *x*_0_ and *y*_0_ can be considered almost zero due to the fact that the proposed antenna is symmetric to both the *XOZ* plane and the *YOZ* plane. Taking the *XOZ* plane as an example, $z_{0}$ can be obtained by Equations (7)–(9).
(7)$$\Psi (\theta )=\vec{k}\cdot {\vec{r}}_{0}+\Psi_{0}$$
(8)$${\vec{r}}_{0}=x_{0}\hat{x}+y_{0}\hat{y}+z_{0}\hat{z}$$
(9)$$z_{0}=\frac{c_{0}}{\pi^{2}f}\int_{0}^{\pi }\Psi (\theta )\cos\theta d\theta$$
where Ψ(θ) is the measured far-field phase pattern, $c_{0}$ is the speed of light, and f is the working frequency.

The optimal position of the proposed antenna, which is selected based on the equivalent phase centers at all frequency points, is determined to be 18.5 mm in front of the feed point of LOQRH. At 2 GHz, the phase center of the proposed antenna and the antenna in [30] deviates farthest from the optimal position. The deviation is 10.7 mm for the proposed antenna and 24.6 mm in [30]. Furthermore, compared to the antenna referenced in [30] at 2 GHz, the stabilization of the phase center in our proposed antenna contributes to an enhanced success rate in direction finding for incoming signals.

## 3. Measurement and Analysis

The radiation characteristics of the proposed antenna are measured by the Anritsu MS46322A VNA (which is manufactured by Anritsu Corporation, headquartered in Atsugi, Kanagawa, Japan) in the anechoic chamber. A broadband horn antenna (A-INFO LB-10180) produced by A-INFO Inc., based in Chengdu, China, was used as the probe antenna to perform far-field measurements of the main polarization and cross-polarization of the LOQRH antenna.

Laser alignment was employed to ensure proper main polarization alignment, and the LOQRH antenna mounted on a turntable was rotated 360 degrees to obtain the radiation pattern for the main polarization. Simultaneously, the broadband horn antenna was rotated by 90 degrees, and the turntable was similarly rotated 360 degrees to capture the cross-polarization radiation pattern. Figure 7 shows the simulated and measured radiation patterns between 2 and 18 GHz. As seen in Figure 7, measured patterns have good symmetry, showing that the CMC has been well suppressed.

Table 2 shows the values of the simulated and measured phase fluctuations in the *XOZ* plane and the *YOZ* plane at some frequencies when taking the optimal position, 18.5 mm in front of the feed point. During the test, we normalized the measured phase values by the measured phase value at 0° to avoid unnecessary slight changes.

If the HPBW of the proposed antenna is less than 90° at a specific frequency point, then we focus on the phase fluctuation within the HPBW; otherwise, we focus on the phase fluctuation within ±45°. It can be seen that within the required range, the phase fluctuation of the proposed antenna within 2–18 GHz does not exceed 22.5° [31] in Table 2, which means that the proposed antenna has a stable phase center within 2–18 GHz.

It can also be found that the measured phase fluctuation is larger than the simulated one in Table 2, which may be caused by the fact that the rotation axis and the phase center of the antenna do not entirely coincide due to the constraints of the measurement conditions.

Figure 8 presents the simulated and measured S parameters and gain. As Figure 8 shows, the measured |S_21_| between the ports is greater than 25 dB, possibly because the two ports were not strictly orthogonal during the test but still met the requirement. The measured |S_11_| is less than −6 dB from 2 to 18 GHz. Affected by the resistors, the antenna’s gain is low when working at 2 GHz. As the frequency increases, the proposed antenna becomes less affected by the resistors, and its gain rises accordingly.

Figure 9 shows the difference between the simulated and measured phase center change in the full space.

The average of the *XOZ* plane and *YOZ* plane phase centers is usually used as the full-space phase center. As shown in Figure 9, the measured full-space phase center change fluctuates less than 20.3 mm in the working frequency band. If 18.5 mm is used as the optimal phase center, the maximum phase center change is 10.7 mm at 2 GHz.

Figure 10 shows a prototype of the proposed antenna.

Table 3 compares the proposed antenna with previously published works. It can be observed that the proposed design achieves miniaturization while maintaining the stability of the phase center and a wide operating bandwidth.

## 4. Conclusions

Based on CMC suppression, a miniaturized, broadband LOQRH antenna for interferometric direction-finding systems is proposed in this article. By introducing the self-balanced feeding structure, the proposed LOQRH can not only achieve small dimensions of 0.27$\;\lambda_{L}$ × 0.27${\;\lambda }_{L}\times$ 0.33$\;\lambda_{L}$ but also effectively suppress the CMC and stabilize the phase center. The measured results illustrate that the proposed antenna has a 2 to 18 GHz bandwidth. Within the working bandwidth, the measured phase center of the proposed LOQRH fluctuates less than 20.3 mm, while the simulated phase center fluctuation is only 14.6 mm. Therefore, the proposed antenna is very suitable for interferometric direction-finding systems, especially for those tiny, multiprobe ones.

## Figures and Tables

**Figure 1 micromachines-16-00044-f001:**
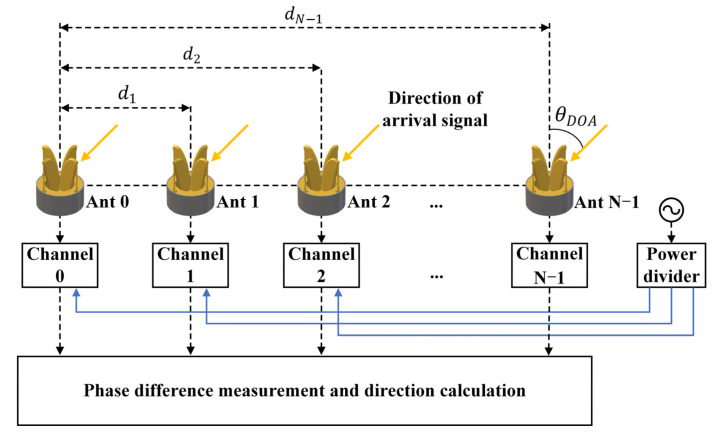
Simplified schematic of an interferometric system.

**Figure 2 micromachines-16-00044-f002:**
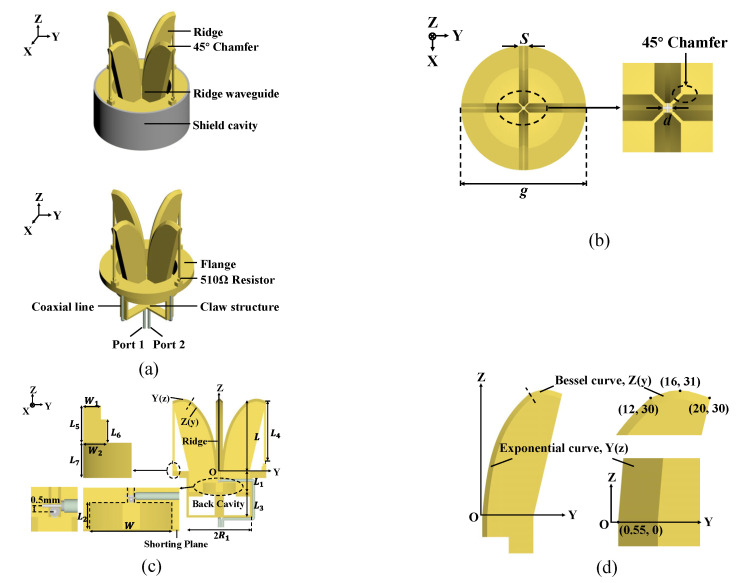
Geometry of the proposed antenna. (**a**) 3D view. (**b**) Top view. (**c**) Section view. (**d**) Detail of the ridge. *S* = 3 mm, *L* = 30 mm, *L*_1_ = 9 mm, *L*_2_ = 4 mm, *L*_3_ = 10 mm, *L*_4_ = 26 mm, *L*_5_ = 3.2 mm, *L*_6_ = 2 mm, *L*_7_ = 3 mm, *R*_1_ = 13 mm, *h*_r_ = 8 mm, *W* = 14 mm, *W*_1_ = 1.5 mm, *W*_2_ = 2 mm, *g* = 40 mm, *d* = 1.1 mm.

**Figure 3 micromachines-16-00044-f003:**
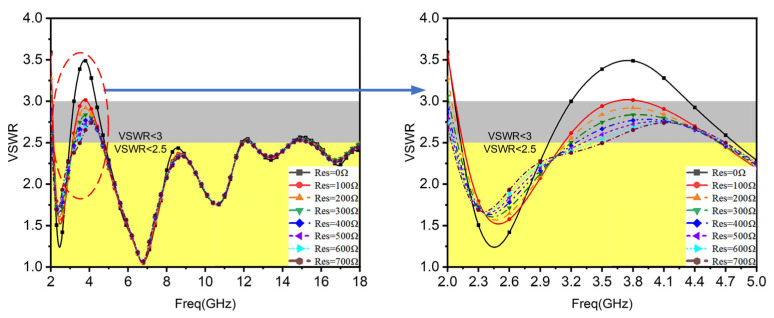
Comparison of VSWR performance of antennas before and after loading resistors.

**Figure 4 micromachines-16-00044-f004:**
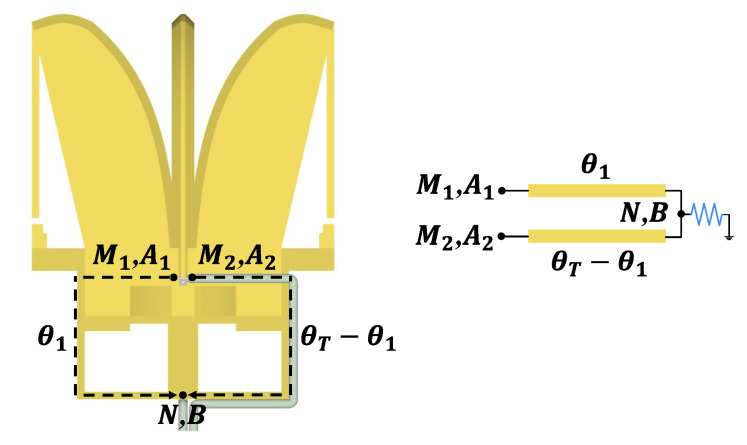
The equivalent odd and even mode analysis model of the self-balanced feeding structure.

**Figure 5 micromachines-16-00044-f005:**
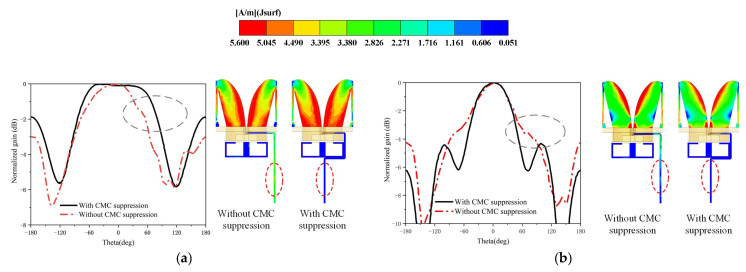
Normalized radiation patterns and surface current distribution of the proposed antenna with or without CMC suppression in the XOZ plane. (**a**) 2 GHz. (**b**) 4 GHz.

**Figure 6 micromachines-16-00044-f006:**
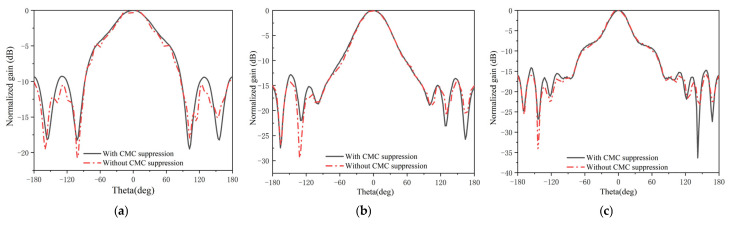
The effect of the CMC suppression at high frequencies. (**a**) 6 GHz, (**b**) 10 GHz, (**c**) 14 GHz.

**Figure 7 micromachines-16-00044-f007:**
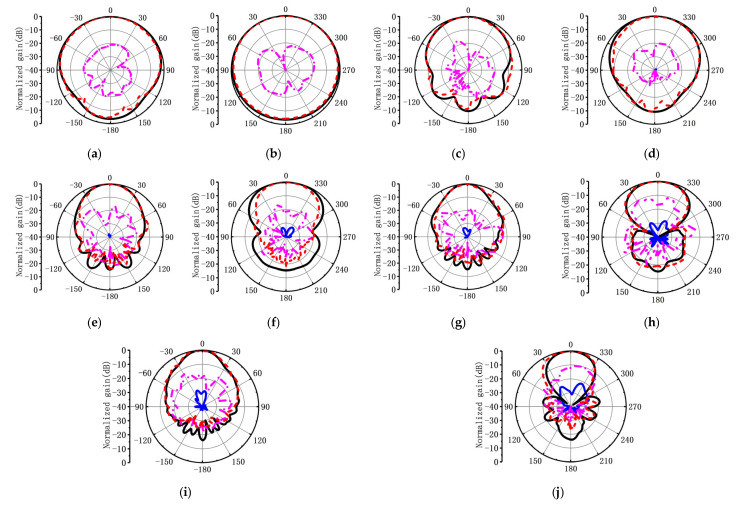
Simulated and measured patterns of the proposed antenna. (**a**) *XOZ* plane at 2 GHz. (**b**) *YOZ* plane at 2 GHz. (**c**) *XOZ* plane at 6 GHz. (**d**) *YOZ* plane at 6 GHz. (**e**) *XOZ* plane at 10 GHz. (**f**) *YOZ* plane at 10 GHz. (**g**) *XOZ* plane at 14 GHz. (**h**) *YOZ* plane at 14 GHz. (**i**) *XOZ* plane at 18 GHz. (**j**) *YOZ* plane at 18 GHz.

**Figure 8 micromachines-16-00044-f008:**
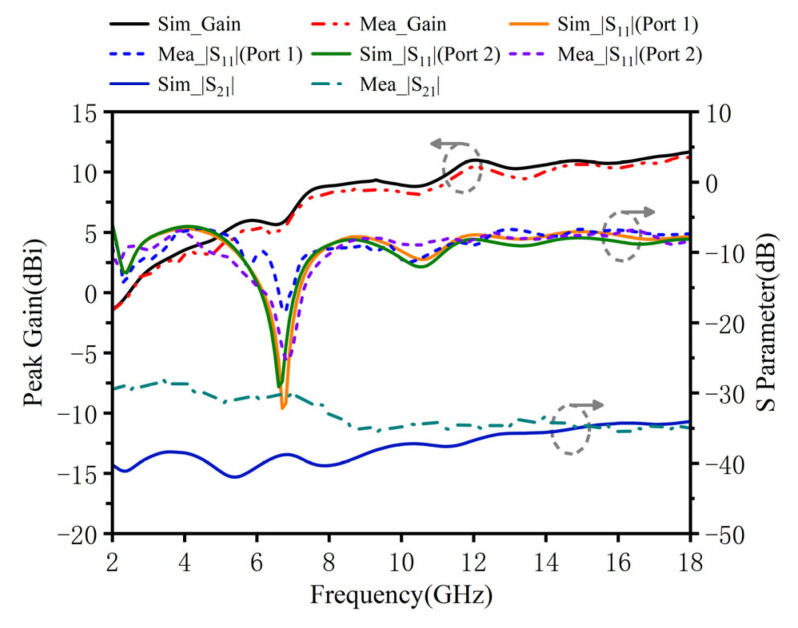
Simulated and measured S parameter and peak gain of the proposed antenna. The direction of the arrows indicates which y-axis is referenced for each data set.

**Figure 9 micromachines-16-00044-f009:**
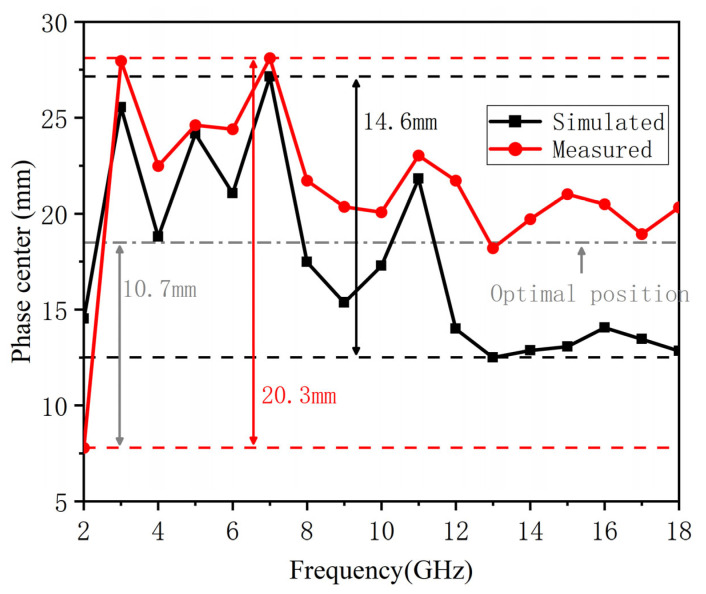
Simulated and measured phase center in the entire space.

**Figure 10 micromachines-16-00044-f010:**
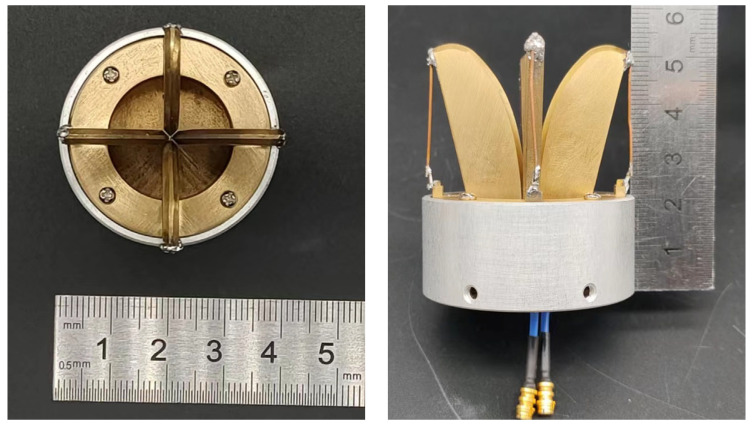
Photographs of the antenna prototype.

**Table 1 micromachines-16-00044-t001:** Symbol table.

Symbol	Explanation
$\theta_{DOA}$	Minimum mean square error solution for the incoming wave angle
$d_{k}$	Length of the interferometer baseline between each unit antenna *N* and the reference antenna 0
$\alpha_{k}$	Ratio of the lengths of the adjacent baseline
$\phi_{\mathrm{W,k}}$	Phase deblurring result of $\phi_{\mathrm{W,k}}$
$\gamma_{\mathrm{Mea,k}}/2\pi \;$	Number of fuzzy baseline k-measurement directions
$\phi_{Mea,k}$	Measured phase difference between channels k and 0
∆ϕ	The phase measurement error
*floor*(·)	The downward rounding function

**Table 2 micromachines-16-00044-t002:** Phase fluctuation values in the *XOZ* plane and *YOZ* plane.

Frequency (GHz)	XOZ Plane Phase Fluctuation (°)		YOZ Plane Phase Fluctuation (°)	
	Simulated	Measured	Simulated	Measured
2	10.2	21.2	7.6	12.0
6	17.4	22.2	11.3	15.8
10	4.3	8.5	18.6	22.5
14	8.4	12.5	12.9	14.5
18	11.4	16.1	14.1	17.8

**Table 3 micromachines-16-00044-t003:** Comparison of main results with previously published works.

Ref. Work	BW (f_max_/f_min_)	Phase Center Change (mm)	Profile (λ_f min_)
[30]	6:1	<30 (2~12 GHz)	1.40 × 1.40 × 1.07
[32]	1.125:1	\	1.61 × 1.61 × 0.30
[33]	2.25:1	\	2.00 × 2.00 × 1.43
[34]	5.2:1	\	1.61 × 1.61 × 1.46
[35]	6:1	<50 (2~12 GHz)	1.14 × 1.14 × 1.04
[36]	3.3:1	<40 (8~18 GHz)	1.29 × 1.29 × 3.16
[37]	8:1	<50 (1.5~12 GHz)	0.42 × 1.42 × 1.32
This work	9:1	<20.3 (2~18 GHz)	0.27 × 0.27 × 0.33

## Data Availability

All data generated or analyzed during this study are included in this manuscript. There are no additional data or datasets beyond what is presented in the manuscript.

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
