# Peer review of "A Miniaturized Loaded Open-Boundary Quad-Ridge Horn with a Stable Phase Center for Interferometric Direction-Finding Systems"

_micromachines, 2024, doi:10.3390/mi16010044_

Round 1
Reviewer 1 Report
Comments and Suggestions for Authors
This paper proposes a miniaturized loaded open-boundary quad-ridge horn antenna with a stable phase center. However, there are still some problems should be addressed. Here are my comments.
1.The miniaturization of the antenna is one of the innovations of the work, it is better to show the miniaturization design process and the effect of loading the resistance between the ridge and the fixed flange.
2.The CMC suppression effect is frequency related, but Fig. 4 only shows the normalized radiation patterns and current distributions at 2 GHz and 4 GHz, which does not give an indication of the effect over the whole bandwidth. I think it would be better to show the effect of the CMC suppression structure at high frequencies.
3.From Fig. 6, the peak gain of the antenna at 2 GHz is less than 0 dBi, and in general the band of the antenna may not be able to cover 2 GHz. Is this acceptable and if so, please give an explanation.
4.Fig. 7 seems to show the phase center, but the vertical legend is “phase center change”, please check it.
Reviewer 2 Report
Comments and Suggestions for Authors
Abstract:
it is too short and they should clearly mentions and explain the problem and challenge they are trying to fix.
Introduction:
it is good to get to the point, but the author at least should present the other possible estimations like TOA, frequency, velocity, phase, ... but they directly jumped to DOA estimation. Besides, they way they start the introduction is vague for people who have little knowledge on these.
Please rearrange the introduction, I suggust that they should syllogism. start from all proper aspect then get to the point
please paraphrase the contribution of the paper at the end of the introduction for better understanding.
section 2 is clear. however, the principles equation are not neccassery.
authors should add their measurement setup for their Co and Cross radiation measurements and explain it entirely.
I suggust that the authors either add a table of comparision with the existing techniques at the end of the introduction or compre with more works in Table 3.
Round 2
Reviewer 2 Report
Comments and Suggestions for Authors
no comments.
Comments on the Quality of English Languagejust need to be slightly improved.